# The Role of Extracellular Vesicles in the Pathogenesis of Hematological Malignancies: Interaction with Tumor Microenvironment; a Potential Biomarker and Targeted Therapy

**DOI:** 10.3390/biom13060897

**Published:** 2023-05-27

**Authors:** Kaushik Das, Tanmoy Mukherjee, Prem Shankar

**Affiliations:** 1Department of Cellular and Molecular Biology, The University of Texas at Tyler Health Science Center, Tyler, TX 75708, USA; 2Department of Pulmonary Immunology, The University of Texas at Tyler Health Science Center, Tyler, TX 75708, USA; tanmoy.mukherjee@uttyler.edu (T.M.); or premshankar.1506@gmail.com (P.S.)

**Keywords:** extracellular vesicles, tumor microenvironment, hematological malignancy, immune cells, stromal cells, endothelial cells, extracellular matrix, lymphatic system, biomarker, drug resistance

## Abstract

The tumor microenvironment (TME) plays an important role in the development and progression of hematological malignancies. In recent years, studies have focused on understanding how tumor cells communicate within the TME. In addition to several factors, such as growth factors, cytokines, extracellular matrix (ECM) molecules, etc., a growing body of evidence has indicated that extracellular vesicles (EVs) play a crucial role in the communication of tumor cells within the TME, thereby contributing to the pathogenesis of hematological malignancies. The present review focuses on how EVs derived from tumor cells interact with the cells in the TME, such as immune cells, stromal cells, endothelial cells, and ECM components, and vice versa, in the context of various hematological malignancies. EVs recovered from the body fluids of cancer patients often carry the bioactive molecules of the originating cells and hence can be considered new predictive biomarkers for specific types of cancer, thereby also acting as potential therapeutic targets. Here, we discuss how EVs influence hematological tumor progression via tumor–host crosstalk and their use as biomarkers for hematological malignancies, thereby benefiting the development of potential therapeutic targets.

## 1. Introduction

Intercellular communication, an essential biological process of multicellular organisms, is mediated by three different mechanisms: (1) cytoplasmic bridges; (2) direct interactions between adjacent cells via membrane proteins; and (3) cellular secretary molecules [1,2]. Recently, a fourth mechanism has been discovered, which includes the intercellular transfer of extracellular vesicles (EVs) [3,4,5,6,7,8,9,10]. EVs are membrane-bound entities released by almost all types of cells into the extracellular environment [11,12]. EVs are known to transport bioactive molecules, such as proteins, lipids, and nucleic acids in the form of DNA, RNA, miRNA (miR), etc., between cells [13,14]. Detectable levels of EVs are found in nearly all biological fluids, such as blood, urine, synovial fluid, and saliva, and they are even found in the interstitial spaces between cells [12,15,16,17]. Since they are protected from degradation by extracellular proteases and RNases, EVs can be stably stored for long-term use [18]. Depending on their biogenesis, size, release mechanism, content, and function, EVs can be broadly classified into microvesicles, exosomes, and apoptotic bodies (Figure 1) [11,12,19,20].

*Microvesicles (MVs)*. MVs are a type of EV generated by direct outward budding of plasma membrane from cells [13,21]. The formation and release of MVs from cells typically require the interplay of cytoskeletal components, such as actin and microtubules; molecular motors, such as kinesin and myosin; and fusion machinery, such as SNAREs and tethering factors [22,23]. MVs typically ranging from 100 nm to 1 µm in diameter [11,12,15,19,21]. Because MVs are generated from plasma membrane by outward budding, they carry cytosolic and plasma membrane-associated proteins, e.g., proteins clustered at the plasma membrane, such as tetraspanins, which could serve as markers for MVs regardless of the originating cells [24,25]. Other cytoskeletal proteins, such as heat shock proteins, integrins, and proteins containing post-translational modifications, such as glycosylation and phosphorylation, have also been shown to be present in MVs [26,27,28].

*Exosomes.* Exosomes are the other subtype of the EV of endocytic origin [15,29] with a typical diameter of 30 to 150 nm [30,31]. Specifically, exosomes are formed by inward budding of early endosomal membrane, which matures into multivesicular bodies (MVBs) [12,15,32]. MVBs eventually fuse with the plasma membrane, releasing their content of exosomes into the extracellular space [32,33,34,35,36]. The regulation of MVBs and the formation and subsequent release of exosomes are mediated by the endosomal sorting complexes required for transport (ESCRT) pathway [37,38,39]. Since exosome generation is mediated by the ESCRT pathway, ESCRT and its accessory proteins (Alix, TSG101, HSC70, and HSP90β) are believed to be present in all exosomes regardless of the type of originating cells and hence serve as exosome markers [40,41,42,43,44]. Other than the ESCRT pathway, exosome generation is also thought to be dependent on sphingomyelinase enzymes since, in some instances, cells with ESCRT deficiency also produce significant numbers of CD63+ exosomes [45,46,47,48]. Both exosomes and MVs have been shown to participate actively in cell–cell communication, maintenance of cells, and tumor progression by transporting their cargo between cells [49,50]. EVs are readily taken up by the recipient cells, either by direct fusion with the plasma membrane or by fusion with the endosomal membrane after endocytosis [51,52,53,54].

*Apoptotic bodies.* Apoptotic bodies, which are in general not considered to be a true form of EV, are larger, ranging from 50 nm to 5 µm in diameter [55,56,57], and are released from cells undergoing programmed cell death [19,56]. These bodies are generated by the separation of plasma membrane from the cytoskeleton due to enhanced hydrostatic pressure during contraction of cells [58,59]. In contrast with MVs and exosomes, apoptotic bodies contain cellular organelles, chromatin, and a few glycosylated proteins [19,42,60,61]. Hence, higher levels of nuclear proteins (such as histones), mitochondrial proteins (such as HSP60), Golgi, and endoplasmic reticulum-associated proteins (such as GRP78) are expected to be observed in apoptotic bodies.

## 2. Tumor-Derived EVs

Tumor-derived EVs are distinguished from normal cell-secreted EVs due to the presence of unique tumor-specific ‘labels’ [62,63,64,65]. Tumor-derived EVs have been shown to carry oncogenic proteins or nucleic acids (such as DNA, RNA, miRNAs, etc.) which facilitate tumor progression. Oncogenic bioactive molecules are enriched in tumor-derived EVs compared to normal cell-derived EVs [66,67,68]. For example, chromosome segregation 1 like protein (CSE1L), a transmembrane protein, is enriched in tumor-derived EVs, not only triggering Ras-dependent EVs biogenesis but also promoting metastasis of B16F10 and melanoma cells [69]. Adriamycin-resistant breast cancer cell-derived EVs were shown to carry transient receptor potential cation channel subfamily C member 5 (TrpC5) and to transfer of EV. TrpC5 confers endothelial cell resistance against chemotherapeutic regimens [70]. On the other hand, the transfer of oncogenic nucleic acids such as miRNAs, specifically miR-221 from highly aggressive breast tumor cells to nonaggressive cancer cells, via EVs, contributing to the promotion of epithelial-to-mesenchymal transition (EMT) [71] and leading to the induction of proliferation and metastasis while preventing drug-induced apoptosis of EVs’ fused recipient cells [5]. Therefore, tumor cells more often induce oncogenic transformations into normal healthy cells via the transfer of oncogenic bioactive molecules through EVs [72,73].

## 3. Hematological Malignancies

Hematological malignancies are defined as tumors that commence in blood-forming tissues, such as bone marrow or cells of the immune system, resulting in leukemias, lymphomas, and myelomas [74,75]. Hematological malignancies are considered to be among the leading causes of cancer-related deaths worldwide [76,77,78]. In the United States itself, an estimated 184,710 new cases of hematological neoplasms were reported with 57,380 deaths in 2023, and the incidence increases with age [79]. GLOBOCAN 2020 reported non-Hodgkin lymphoma to be the predominant hematological cancer worldwide with 544,352 new cases and 259,793 deaths, followed by leukemia, with 474,519 new cases and 311,594 deaths worldwide [80]. The outbreak of COVID-19 further increased the death rate of patients suffering from various hematological malignancies [81,82,83,84,85].

EVs are known to play an important role in cell–cell communication via the transfer of bioactive cargo molecules. However, the role of EVs in the crosstalk of tumor cells with cells in the tumor microenvironment (TME) and other distant cells remains to be completely determined in the context of the pathogenesis of hematological malignancies. The present review highlights how hematological cancer cells communicate with the cells in the TME and vice versa via the release of EVs. The first part of the review focuses on the modern classification of hematological malignancies, followed by understanding the EVs’ role in cell–cell and cell–extracellular matrix communication in the TME. The final part of the review reveals how EVs could be used as biomarkers for hematological malignancies and their contributions to drug resistance.

## 4. Hematological Malignancies: The Modern Classification

Over the years, different classification schemes have been established to define various hematological malignancies; however, the World Health Organization (WHO) classification is the most prevalent [86]. Based on features such as morphology, lineage, clinical attributes, cytogenetics, and molecular pattern, the WHO 5th edition (WHO5) broadly categorizes hematological neoplasms into myeloid neoplasms, lymphoid neoplasms, myeloid/lymphoid neoplasms with eosinophilia and tyrosine kinase gene fusions, mastocytosis, histiocytic/dendritic neoplasms, and mixed myeloid and lymphoid neoplasms [87,88]. In the present section, we discuss a detailed classification of myeloid- and lymphoid neoplasms, followed by a brief description of other hematological neoplasms (Table 1).

*Myeloid neoplasms.* This type of neoplasm is obtained from bone marrow progenitor cells, which differentiate into megakaryocytes, granulocytes, erythrocytes, and monocytes. Exceptions exist, such as chronic myeloid leukemia (CML), in which the originating cells are pluripotent hematopoietic stem cells that differentiate into lymphoid cells. Myeloid neoplasms are further classified as follows.

(1). Myeloproliferative neoplasms (MPNs): MPNs, a type of clonal stem cell disorder, are caused by genetic mutations, resulting in abnormal activation of pro-growth signaling, ultimately leading to uncontrolled proliferation of myeloid progenitors, such as megakaryocytes, granulocytes, or erythrocytes. MPNs are sub-classified into: (A) *chronic myeloid leukemia (CML)*—caused by excessive proliferation of the granulocyte lineage, leading to increased myeloid maturation, which is associated with the BCR:ABL1 fusion gene; (B) *classic BCR:ABL1-negative MPNs*—these neoplasms lack the BCR:ABL1 fusion gene, which includes polycythemia vera (PV), characterized by increased red blood cells; essential thrombocytopenia (ET), a clonal disorder associated with thrombocytosis; and primary myelofibrosis (PF), caused by a polyclonal increase in fibroblasts leading to bone marrow fibrosis; (C) *chronic neutrophilic leukemia (CNL)*—caused by overproduction of matured granulocytes; (D) *chronic eosinophilic leukemia (CEL)*—caused by clonal proliferation of morphologically abnormal eosinophils and their precursors, resulting in hypereosinophilia; and (E) *myeloproliferative neoplasm, unclassifiable*—these neoplasms are classes of MPNs not satisfying the criteria to categorize into specific MPN types or having overlapping MPN subtype characteristics.

(2). Myelodysplastic neoplasms/syndromes (MDS): These neoplasms are caused by ineffective blood cell production, leading to dysplasia or cytopenia and often resulting in bone marrow failure. In accordance with WHO5, MDS are classified with respect to genetic abnormalities or morphology. According to genetic abnormalities, MDS are subtyped into: (A) *MDS with low blasts and isolated del(5q)*—5q deletion, bone marrow blasts (BMB) < 5% in bone marrow and peripheral blasts (PB) < 2%; and SF3B1 or TP53 mutation; (B) *MDS with low blasts and SF3B1 mutation*—absence of 5q deletion, BMB < 5% and PB < 2%, SF3B1 mutation, and ≥15% ring sideroblasts; and (C) *MDS with biallelic TP53 inactivation*—equal to or more than two TP53 mutations or one mutation with TP53 copy number loss or copy neutrality, resulting in loss of heterozygosity, complex cytogenetics, BMB, and PB < 20%. Morphologically defined MDS can be classified into: (A) *MDS with low blasts*—BMB < 5% and PB < 2%; (B) *MDS hypoblastic*—BM cellularity < 20%; and (C) *MDS with increased blasts (MDS-IB)*—these neoplasms include MDS-IB1, BMB blasts 5–9%, and PB blasts 2–4%; MDS-IB2, BMB blasts 10–19%, and PB blasts 5–19%; MDS with increased blasts and fibrosis; BM fibrosis with BMB blasts 5–19%; and PB blasts 2–19%.

(3). Acute myeloid leukemia (AML): These neoplasms are characterized by highly aggressive and genetically heterogenous myeloid malignancies. They are further classified as: (A) *AML with recurrent genetic abnormalities*—caused by genetic abnormalities and subcategorized into characteristic chromosomal changes due to inversions or reciprocal chromosomal rearrangements and no associated karyotypic abnormalities; (B) *AML with myelodysplasia-related features*—observed in patients with a history of MDS or MDS/myeloproliferative neoplasms (MPNs) and lacking the genetic features of AML; (C). *therapy-related AML*—complications arise after treatment with chemo- or radiation therapy; (D) *AML not otherwise specified (NOS)*—having recognizable clinical, morphologic, and immunophenotypic features but lacking karyotypic and diagnostic features of AML; (E) *myeloid sarcoma*—these neoplasms are masses of tumor blasts outside the bone marrow but effacing into the tissues; and (F) *myeloid proliferations related to Down syndrome (DS)*—occurring in Down syndrome, which is associated with an increased risk of AML.

(4). Myeloid neoplasms with mutated TP53: These neoplasms are new types of myeloid neoplasms, which are further classified as MDS, MDS/AML, and AML depending on TP53 mutations and blast counts. They are categorized into—*MDS with mutated TP53*, <10% blasts; *MDS/AML with mutated TP53*, 10–19% blasts; and *AML with mutated TP53*, >20% blasts.

*Lymphoid neoplasms.* Lymphoid neoplasms are typical types of cancer, affecting cells that normally give rise to B-lymphocytes, such as lymphocytes; plasma cells; and T-lymphocytes, such as Treg, T_H_, and T_C_. Depending on the B-cell and T-cell/NK-cell lineage, lymphoid neoplasms can be classified into precursor lymphoid neoplasms, mature B cell neoplasms, Hodgkin lymphoma, and mature T cell or NK cell lineage neoplasms.

(1). Precursor lymphoid neoplasms: These neoplasms are highly aggressive and are referred to as acute lymphoblastic leukemia/lymphoblastic lymphoma (ALL/LBL) because they evolved from either leukemic or lymphatic cells. They are further classified as—*B-cell ALL/LBL*—arising from precursor B-cells and representing most of ALL/LBLs; and *T cell ALL/LBL*—derived from T-cell precursors comprising 15% all ALL/LBL populations.

(2). Mature B-cell neoplasms: According to WHO5, lymphoid neoplasms of matured B-cells are further classified as: (A) *chronic lymphocytic leukemia (CLL)/small lymphocytic lymphoma (SLL)*—small, matured-appearing lymphocytes; (B) *lymphoplasmacytic lymphoma (LPL)*—derived from post-germinal centers of B-cells; (C) *,monoclonal gammopathies*—a new subtype of lymphoid neoplasms for primary cold agglutinin disease, which are further classified as primary cold agglutinin disease, monoclonal gammopathy of undetermined significance (MGUS) of non-IgM type, or amyloidosis; (D) *plasma cell neoplasms*—related to terminally differentiated germinal centers of B-cells secreting monoclonal IgM; (E) *hairy cell leukemia*—tumors related to the post-germinal centers of B-cells with the appearance of a hairy morphology; (F) *marginal zone lymphoma (MZL)*—neoplasms associated with a marginal zone of matured B-cells; (G) *follicular lymphoma (FL)*—the most common lymphoma, which is associated with the neoplastic germinal center of B-cells and further classified as—classic FL, proliferation of small (centrocytes) and large (centroblasts) neoplastic follicular cells; FL grade 3B, uncontrolled proliferation of centroblasts; and FL with unusual features, comprising FL with blastoid features and FL with predominantly diffuse growth pattern; (H) *mantle cell lymphoma (MCL)*—related to neoplasms of naïve B-cells and antigen-stimulated B-cells; (I) *diffuse large B-cell lymphoma (DLBCL)*—associated with matured B-cells; and (J) *high-grade B-cell lymphomas*—comprising Burkitt lymphoma (BL), a highly aggressive B-cell neoplasm; and other high-grade B-cell lymphomas associated with MYC and BCL-2 rearrangements.

(3). Hodgkin lymphoma (HL): These neoplasms are derived from germinal center and post-germinal center B-cells, comprising the minority population of neoplastic lymphomas and subdivided into: (A) *classic HL (cHL)*—further classified into nodular sclerosis cHL, mixed cellularity cHL, lymphocyte-rich cHL, or lymphocyte-depleted cHL; and (B) *nodular lymphocyte-predominant HL (NLPHL)*—these neoplasms are uncommon, retaining immunophenotypic features of germinal neoplastic B-cells.

(4). Mature T-cell or NK-cell lineage lymphomas: These neoplasms are further categorized into: (A) *peripheral T cell lymphoma (PTCL)*—comprising peripheral T cell lymphoma that does not fit the criteria of T-cell lymphoma, NOS; anaplastic large cell lymphoma (ALCL)—according to the expression of anaplastic lymphoma kinase (ALK), these neoplasms are subcategorized into “ALCL, ALK-positive” or “ALCL, ALK-negative”; and “breast implant-associated ALCL”; and follicular helper T-cell lymphoma—including “angioimmunoblastic T-cell lymphoma (AITL)” and related follicular TH cells; extranodal NK-/T-cell lymphoma, nasal type—these neoplasms are Epstein–Barr virus (EBV)-associated neoplasm arises in the lymph nodes; subcutaneous panniculitis-like T cell lymphoma; hepatosplenic T-cell lymphoma—highly aggressive and related to immunosuppression; and primary intestinal T-cell lymphomas—aggressive T-cell lymphomas in the intestinal tract; (B) *primary cutaneous peripheral T-cell lymphomas*—related to cutaneous peripheral T-cells; (C) *adult T-cell leukemia-lymphoma (ATL)*—associated with peripheral T-cells originating from T-cell leukemia virus (HTLV) type 1-infected CD4+ T-cells; (D) *T-cell large granular lymphocyte (TLGL) leukemia*—developed from clonally expanded large granular T-cells; (E) *T-cell prolymphocytic leukemia*—highly aggressive and comprising medium-sized matured T-cells; (F) *NK-cell large granular lymphocyte (NKLGL) leukemia*—highly aggressive, associated with malignant NK-cells; and (G) *aggressive NK-cell leukemia*—considered a nasal type since more often shown to be associated with EBV infection.

*Myeloid/lymphoid neoplasms with eosinophilia and tyrosine kinase (TK) gene fusions.* Eosinophilia is defined as an increase in eosinophil count in the peripheral blood and tissues. More often, myeloid/lymphoid neoplasms are shown to be associated with hyper-eosinophilia (HE), which is defined as an absolute eosinophil count (AEC) >1.5 × 10^9^/L in the peripheral blood and hypereosinophilic syndrome (HES) in a situation of HE in which eosinophil-mediated organ damage or dysfunction is observed. According to the pathogenic mechanisms leading to the expansion of eosinophils, HES are subclassified as *primary*, *secondary*, and *idiopathic*. According to the gene fusion and rearrangements of PDGFRA, PDGFRB, JAK2, FGFR1, ETV6:ABL1, and FLT3, WHO5 defined these types of neoplasms as a separate class. A brief classification of these hematological neoplasms is shown in Table 1.

*Mastocytosis.* These neoplasms are rare, heterogenous forms of neoplasms, resulting from neoplastic mast cells in tissues and various organs. Constitutive activation of KIT receptor is shown to contribute to this form of hematological neoplasm. WHO5 classified this type of neoplasm into *cutaneous mastocytosis (CM)*, in which the disease is limited to skin only; and *systemic mastocytosis (SM)*, involving extracutaneous organ infiltration. A brief discussion is provided in Table 1.

*Histiocytic/dendritic neoplasms.* This type of neoplasm is related to cells that have developed into antigen-presenting dendritic cells and tissue macrophages, further classified in Table 1. The MAPK pathway has been shown to play an important role in histiocytic/dendritic neoplasms.

*Mixed myeloid and lymphoid neoplasms.* Sometimes, neoplasms express the markers of myeloid and lymphoid cells or none, with examples being “*acute undifferentiated leukemia*” and “*mixed phenotype acute leukemia (MPAL)*.” A detailed classification is shown in Table 1.

## 5. EVs in Cell-Cell and Cell-Extracellular Matrix Communication in the TME

The TME is the environment surrounding the tumor cells in the body [89,90]. It consists of immune cells, stromal cells, fibroblasts, extracellular matrix (ECM), and cells of the blood and lymphatic vessels [89,90]. Tumor cells and their TME are in constant interaction, thereby regulating each other either positively or negatively [91]. Dynamic interaction between cancer cells and TME components not only supports tumor growth and development [92,93] but also promotes local invasion and metastatic dissemination of cancer [94,95]. In hypoxic and acidic conditions, the TME often promotes angiogenesis, a process of restoring nutrient and oxygen supply, as well as removing metabolic waste [96,97,98]. Additionally, the infiltration of various immune cells into the TME performs various pro- and anti-tumorigenic functions [99,100,101,102]. EVs play an important role in intercellular communication via the transfer of bioactive cargo molecules between cells [103]. Tumor-derived EVs also act as a communicating vehicle between cancer cells and cells in the TME and in some instances also with distant cells. On the other hand, cells in the TME often release EVs that interact with tumor cells, influencing tumor development and progression.

*Effects of hematological malignancy-derived EVs on immune cells.* Immune cells play a major role in the elimination of tumors through diverse mechanisms, and the evasion of immune surveillance serves as an important step for developing tumor niches and successful establishment of tumors. Immune evasion, a strategy facilitated by tumor-derived EVs is utilized in different ways to target various immune cells (Figure 2). Tumor-derived EVs, through their receptor-mediated uptake, can introduce several suppressive factors, e.g., miRNA, DNA, pro-apoptotic factors, metabolites, and various enzymes, into immune cells (Table 2). They can also alter the activation of immune cells through inhibitory cell surface receptors [104,105,106,107].

In activated T-cells, tumor-derived EVs induce the down-regulation of CD3ζ and JAK3 expression via transcriptional regulation, thereby facilitating Fas/FasL-mediated apoptosis of CD8(+) T-cells [108]. Chronic lymphocytic leukemia (CLL)-derived EVs were shown to down-regulate CD69 expression in T-cells via miR-363 transfer, thereby affecting effector T-cell migration [109,110,111]. Tumor cells are often shown to evade the host immune system via the activation of the PD-L1/PD-1 pathway. PD-L1 is expressed on the surfaces of various tumor cells, whereas its receptor, PD-1, is present on T-cells. PD-L1 binding to PD-1 results in the apoptosis of T-cells, thereby evading host immune responses. The upregulation of PD-1 has been observed in various T-cell populations after exposure to diffuse large B cell lymphoma (DLBCL)-derived EVs [112]. EVs released from B-cell lymphoma (BCL) under chemotherapy are enriched with CD39 and CD73, and they hydrolyze ATP, which is generated from chemotherapy-treated tumor cells and transformed into adenosine [113], which in turn affects the immune system by inhibiting T-cell activity and proliferation [114].

**Table 2 biomolecules-13-00897-t002:** The effect of EVs, derived from hematological malignancy on immune cell function.

Originating Cells	Effector Cells	Malignancies	Functions	References
Leukemic cells	CD8(+)T-cells	Solid tumors or AML	Down-regulates CD3ζ and JAK3 expression and promotes Fas/FasL-mediated T-cell apoptosis	[108]
Leukemic cells	CD4(+)T-cells	CLL	Down-regulates CD69 expression via miR-363 transfer and affects effector T-cell migration	[109,110,111]
Lymphoma cells	T-cells	DLBCL	Induces PD-1 expression in T-cells and enhances T-cell apoptosis	[112]
Lymphoma cells	T-cells	BCL	Carries CD39 and CD73 and hydrolyzes ATP to generate adenosine to inhibit T-cell activity and proliferation	[114]
Lymphoma cells	NK-cells, APCs	BCL, TCL	Carries MHC, APO2L, FASL, TCR, and NKG2D and inhibits NK-cells cytotoxicity and antigen processing of APCs	[115,116,117]
Non-leukemic cells	NK-cells	CLL	Carries BAG6 and activates NK-cells, but activated NK-cells are eliminated by lymphocytes	[118]
Myeloma cells	MDSCs,	MM	Induces growth and immunosuppressive activity	[119,120]
	NK-cells,		Reduces NK-cells’ cytotoxicity	[121]
	Immune cells		Carries ectoenzyme, CD38, which converts nucleotides into adenosine to suppress immune system	[122]
Leukemic cells	NK-cells	AML	EVs’-bound TGFβ1 reduces NK-cells’ cytotoxicity	[123]
Lymphoma cells	Monocytes	-	Releases TNF-α, IL-1β, and IL-6 and prevents monocyte differentiation into dendritic cells	[124]
Lymphoma cells	Macrophages	DLBCL	Transfers MyD88 and stimulates pro-inflammatory NF-κB signaling pathway	[125]
Leukemic cells	Macrophages	CML	Polarizes macrophages to M2-phenotype to induce TNF-α and IL-10 expression and down-regulates NO and ROS generation	[126]
Tumor cells	Neutrophils	CAT	Promotes NET formation, reduced generation of suppressor cells	[127,128,129]
Lymphoma cells	MDSCs	Lymphoma	Carries HSP72 and promotes suppressive functions	[124]

*Abbreviations of malignancies:* AML, acute myeloid leukemia; CLL, chronic lymphocytic leukemia; DLBCL, diffuse large B cell lymphoma; BCL, B-cell lymphoma; TCL, T-cell lymphoma; MM, multiple myeloma; CAT, cancer associated thrombosis; CML, chronic myelogenous leukemia.

EVs are known to carry surface receptors of the originating cells. For example, EVs from B- and T-cell lymphomas are capable of carrying cell surface molecules, such as major histocompatibility complex (MHC), Apo2 ligand (APO2L), Fas ligand (FASL), T-cell receptor (TCR), and natural-killer group-2 member-D (NKG2D), which not only inhibit the cytotoxicity of NK-cells, thereby promoting T-cell apoptosis, but also down-regulate the processing of antigens by antigen presenting cells (APCs) [115,116,117]. The plasma soluble ligand, BAG6, in CLL patients binds to the receptor NKp30 on NK-cells, causing NK-cell inactivation [118]. In contrast, BAG6, carried by the EVs, activates NK cells, which have the ability to kill tumor cells [118]. Hence, the dysregulated balance between the soluble form of BAG6 and its EVs’ bound form determines the immune evasion of CLL. EVs from multiple myeloma (MM) cells suppress the immune system through different mechanisms. First, MM-derived EVs enhance the growth and immunosuppressive activity of myeloid-derived suppressor cells (MDSCs) in both in vitro and in vivo MM xenograft murine models [119,120]. Second, EVs from MM cells reduce NK-cells’ cytotoxic activity [121]. Third, the ectoenzyme, CD38, on MM-derived EVs converts nucleotides into adenosine, which is a well-known suppressor of the immune system [122].Finally, CD38-positive EVs were shown to be internalized by FcR-positive cells, such as monocytes, MDSCs, and NK cells, after binding to an anti-CD38 mAb (daratumumab), although the effects are still under investigation [123]. TGFβ1, on the other hand, was shown to play an important role in immune-evasive mechanisms of leukemic EVs [130]. EVs from the sera of acute myeloid leukemia (AML) patients are enriched with membrane-bound TGFβ1, which significantly reduces the killing properties of NK-cells [131]. Moreover, TGFβ1 on chronic myelogenous leukemia (CML) EVs stimulates the proliferation and colony formation of CML cells [130].

Monocytes are the highly dynamic cells differentiated into macrophages and dendritic cells to effectively protect the body from tumor assault. The fusion of lymphoma-derived EVs with monocytes results in the release of TNF-α, IL-1β, and IL-6, which in turn impair monocytic differentiation into dendritic cells [123]. DLBCL-derived EVs readily transfer MyD88 to the macrophages, thereby stimulating the pro-inflammatory signaling pathway, NF-κB, independent of TLR and IL-1R activation [125]. Similarly, CML-derived EVs have been demonstrated to alter the macrophage polarization to a more M2 phenotype within the tumor microenvironment, hence demonstrating up-regulation of IL-10 and TNF-α expression while down-regulating macrophage NO and ROS generation [126]. More often, in a tumor environment, M2 macrophages are converted into tumor-associated macrophages (TAMs) with the potential of releasing pro-tumorigenic growth factors, cytokines, and chemokines to enhance tumor progression [132,133,134,135].

Like monocytes, granulocytes, such as neutrophils, are highly plastic cells that can be readily influenced by tumor-derived EVs. Tumor-derived EV-treated neutrophils have been shown to promote NET formation and reduce the generation of suppressor cells, which are beneficial for tumor progression [127,128,129].

MDSCs, the primary cells associated with reducing the tumorigenicity of T-cells and NK-cells, have been shown to be modulated by heat shock proteins (HSPs) inside the EVs, such as HSP70 and HSP72 [124,136,137].

*Crosstalk between tumor cells and stromal cells through EVs.* Stromal cells refer to a highly heterogenous population of cells that provide structural and physiological support for hematopoietic cells. In cancer, stromal cells often contribute to disease progression by supporting growth, development, and metastasis of tumors (Figure 3). Table 3 also summarizes interactions between tumor cells with cells in the TME via the release of EVs.

In CLL patients, activation of the tumor microenvironment, thereby promoting disease progression, is favored by tumor cell-secreted EVs [138]. CLL-derived EVs activate the AKT survival pathway in stromal cells, in turn releasing vascular endothelial growth factor (VEGF), thereby contributing to tumor survival [139]. Again, CLL-derived EVs facilitate the conversion of stromal cells into cancer-associated fibroblasts (CAFs), promoting tumor metastasis and angiogenesis [140,141].

In acute lymphoblastic leukemia (ALL), EV-mediated transfer of galectin 3 (GAL3) from stromal cells to ALL cells stimulates endogenous GAL3 expression, which confers protection against drug treatment [142]. On the other hand, EVs derived from ALL cells induce a metabolic shift from oxidative phosphorylation to aerobic glycolysis in stromal cells [143]. 

**Table 3 biomolecules-13-00897-t003:** Tumor cells communicate with the stromal cells and vice versa via EVs transfer.

Originating Cells	Effector Cells	Malignancies	Functions	References
Tumor cells	Stromal cells	CLL	Tumor-derived EVs induce stromal cells to release VEGF to promote tumor survival	[139]
Tumor cells	Stromal cells	CLL	Tumor-derived EVs convert stromal cells into CAFs, thereby promoting metastasis and angiogenesis	[140,141]
Stromal cells	Cancer cells	ALL	Stromal cell-derived EVs induce GAL3 expression in cancer cells, hence inducing drug resistance	[142]
Tumor cells	Stromal cells	ALL	Switch oxidative phosphorylation to aerobic glycolysis in favor of cancer progression	[143]
Stromal cells	Tumor cells	MM	Stromal cells from MM induce proliferation, migration, and survival of tumor cells	[144]
Tumor cells	Endothelial cells	MM	Tumor cell-derived EVs in hypoxic conditions affect miR-135b, targeting HIF-1 pathway to promote angiogenesis	[145]
Tumor cells	Endothelial cells	MM	Tumor cell-secreted EVs activate endothelial STAT3 pathway, which promotes angiogenesis and tumor growth	[146]
Tumor cells	Osteoclasts	MM	MM-derived EVs support osteoclast growth and migration	[147]
Fibroblasts	Tumor cells	MM	EVs carry clBcl-xL, which helps in EV uptake by the tumor cells to promote tumor proliferation	[148]
Tumor cells	MSCs	ATL	Tumor cell-derived EVs transfer miR-155 and miR-21 to the MSCs, thereby inducing MSC proliferation	[149]
Leukemic CD34+ cells	MSCs	AML	AML CD34+ cell-derived EVs reduce further development of CD34+ cells from MSCs via miR-7977	[150]
MSCs	CD34+ cells	MPN	EVs carry miR-155 from MSCs, increasing granulocyte CFU numbers in neoplastic CD34+ cells	[151]
Leukemic cells	MSCs	CML	CML-derived EVs induce IL-8 release from MSCs, thereby promoting CML survival	[152]
BCR-ABL + tumor cells	Mononuclear	CML	Induce genome instability, leading to malignant transformation of cells	[153]
Tumor cells	Fibroblasts	TCL, CML	Transfer of hTERT mRNA via the EVs results in induced hTERT expression in the fibroblasts, leading to genome instability	[154,155]

*Abbreviations of malignancies:* CLL, chronic lymphocytic leukemia; ALL, acute lymphocytic leukemia; MM, multiple myeloma; ATL, adult T-cell leukemia/lymphoma; AML, acute myeloid leukemia; MPN, myeloproliferative neoplasm; CML, chronic myelogenous leukemia, TCL, T-cell leukemia.

In the case of MM, stromal cell-derived EVs from MM induce proliferation, migration, and survival of tumor cells, whereas normal stromal cell-secreted EVs prevent tumor proliferation [144]. Under hypoxic conditions, MM-derived EVs were shown to affect miR-135b, which targets the HIF-1 pathway to induce tumor angiogenesis [145]. In another study, MM-derived EVs were reported to modulate the STAT3 pathway in endothelial cells, not only promoting angiogenesis but also inducing tumor growth via the release of VEGF and IL-6, respectively [146]. Moreover, EVs from MM are also capable of inducing growth and migration of osteoclasts (OCs), thus promoting the development of bone diseases in MM patients [147]. Stromal fibroblast-derived EVs actively carry Bcl-xL and its cleaved counterpart, clBcl-xL, facilitating the uptake of EVs by MM cells, hence promoting tumor proliferation [148]. 

Tumor cells from adult T-cell leukemia/lymphoma (ATL) often release miR-155 and miR-21 through EVs, triggering mesenchymal stem cell (MSC) proliferation and aiding the development of a friendly environment for leukemic progression [149]. Another study indicated that AML CD34+ cell-derived EVs efficiently transfer miR-7977 to MSCs to reduce proliferation of CD34+ cells [150].

miR-155 is found to be selectively packaged in the EVs of MSCs from myeloproliferative neoplasms (MPNs), increasing granulocyte CFU numbers in neoplastic CD34+ cells [151]. Moreover, leukemic cell survival in CML is enhanced by IL-8 production from MSCs upon incorporation of CML cell-derived EVs [152]. Another study showed that EVs from BCR-ABL+ CML tumor cells induced genomic instability in normal mononuclear cells, leading to malignant transformations [153]. Additionally, hTERT mRNA is known to be transported from TCL and CML cells to fibroblasts via EVs, leading to ectopic hTERT expression in fibroblasts [154] and a resultant switch to a tumor-like phenotype [155]. 

*Interaction of EVs with the endothelium in the context of hematological malignancies*. Hematological neoplasm-derived EVs are believed to communicate with the cells of the endothelium and vice versa. Hematological tumors, endowed with angiogenic-promoting ability, are dependent on the vascular endothelium for growth, migration, and invasion [156]. More often, tumor-EVs have been shown to activate the endothelial cells, contributing to tumor angiogenesis, whereas endothelial EVs in the TME induce tumor cells to remodel ECM components, thereby providing supplements for the growth of the tumor [157]. For example, EV-mediated transfer of VEGF and VEGF receptor from AML to endothelial cells promotes endothelial glycolysis, leading to vascular remodeling and chemoresistance [158,159]. Multiple signaling pathways appear to be regulated in MM, leading to increased viability of a bone marrow endothelial cell line (STR10), enhanced, angiogenesis and immunosuppression, further facilitating the progression of MM [120,159]. Piwi-interacting RNA-823 (piRNA-823), essentially carried by MM-derived EVs, transforms the endothelial cells, in turn promoting MM growth [159,160]. However, treatment of MM with a protease inhibitor, bortezomib, bestowed anti-angiogenic properties on MM-derived EVs [161]. CML-derived EVs induce the expression of VCAM-1 and IL-8 in endothelial cells, thereby promoting angiogenesis [162]. However, pretreatment of CML cells with curcumin decreases endothelial proliferation and migration via the release of miR-21-enriched EVs [163]. Moreover, CML-EVs, packaged with miR-126, have been found to negatively regulate the expression of VCAM-1 and CXCL12 in endothelial cells, thereby restricting CML motility and adhesion [159,164]. In CLL, tumor-derived EVs are shown to up-regulate the expression of ICAM-1, CXCL1, and IL-34 in endothelial cells, leading to angiogenesis and CLL proliferation [140]. In PML, EVs carry a significant amount of retinoic acid receptor-α (RAR-α), and the transfer of PML-EVs’ RAR-α to the endothelial cells results in the acquisition of tissue factor, hence imparting pro-coagulant properties to the endothelial cells [159,165].

*Interaction of EVs with extracellular matrix (ECM)*. ECM, the physical scaffold, plays a pivotal role not only in the communication of cells with nearby cells but also in the growth, function, and movement of cells [166,167,168]. Tumor cells often communicate with the ECM via the release of EVs [169,170]. For example, surface heparan sulfate (sHS)-positive EVs from MM cells bind to one ECM component, fibronectin, thereby acting as a ligand for sHS-positive target cells. The binding of EVs’ fibronectin with sHS-positive cells triggers the activation of MAPKs (p38 and ERK1/2), resulting in the production of MMP-9 and DKK1, which essentially regulate the invasion of MM cells [171]. Cancer cell-derived EVs often induce the secretion of EMMPRIN, MMP-9, and IL-6 from human monocytic cell lines, thereby modulating the ECM to promote migration and inflammation, ultimately leading to tumor progression [172]. CD30+ EVs have often been shown to stick to long actin or tubulin protrusions of HL cells grown in 3D-matrigel or tissues, which could be a guiding mechanism of EVs to reach distant cells, thereby enabling cell–cell communication [173]. Figure 4 illustrates the contribution of EVs to the progression of hematological tumors by interacting with the ECM.

*EVs in lymphatic malignancies.* Lymphoma is a hematological malignancy associated with the lymphatic system [174,175]. A growing body of evidence has indicated that EVs actively participate in the pathogenesis of various lymphomas [176,177]. The incorporation of EVs, generated from EBV-associated lymphomas into monocytes or macrophages, transforms the immune regulatory mechanisms, leading to tumor evasion [178]. Lymphoma-derived EVs have often been shown to promote the angiogenic process by delivering angiogenic mRNA, miRNA, and proteins such as VEGF [179]. EBV lymphoma-induced macrophages have been shown to release secreted phospholipase A2 of group X (sPLA2-X), which hydrolyzes lymphoma-derived EVs’ phospholipids, thereby allowing for better uptake of EVs and associated lipid mediator signaling in TAMs, contributing to lymphoma growth [180]. HL-derived EVs were demonstrated to be internalized by TME fibroblasts and promote the release of pro-inflammatory cytokines, growth factors, and angiogenic factors, which together contribute to growth of lymphomas [181]. Moreover, HL-derived large EVs were shown to promote the release of IL-1β from monocytes depending on CD44 transfer, whereas both large and small HL-EVs confer immunomodulatory effects through eATP [182]. A recent study indicated that plasma EVs of pediatric HL can be used as a potential biomarker for relapse occurrence of HL [183]. DLBCL-derived EVs were found to be taken up by the tonsillar cells and stromal cells, which contribute to the progression of DLBCL [184]. Moreover, the PD-L1+ EV population was shown to be elevated in the plasma of DLBCL patients and could serve as a biomarker for DLBCL [185]. A recent study also indicated that DLBCL-tumor-derived EVs carry miR-125b-5p, which is readily taken up by DLBCL cells and reduces DLBCL sensitivity to rituximab via miR-125b-5p-mediated targeting of TNFAIP3, reflecting the ability of DLBCL-EVs to influence other cells in the TME [186]. Burkitt lymphoma-derived EVs not only inhibit autophagy and apoptosis but also promote lymphoma growth via miR-106a-mediated targeting of Beclin1 [187].

## 6. EVs: The Biomarker of Hematological Malignancies: Contribution in Drug Resistance

*EVs as biomarkers*. Biomarker identification, expressed differentially in different cell populations, as well as patients, is of prime importance for the diagnostic, prognostic, and therapeutic relevance of cancer. EVs carrying bioactive molecules such as proteins, DNA, RNA, miRNAs, etc., of the originating cells serve as excellent biological markers for the detection of cancer [188]. The release of EVs has been shown to be increased significantly during the development of cancer; hence, EVs in the body fluids of cancer patients serve as prognostic biomarkers for cancer [189,190,191]. The half-life of an EV-based biomarker is relatively longer since the content of the EVs is protected from extracellular proteases and nucleases, contributing to biomarker stability [192]. More often, at the disease stage, response to therapy, tumor burden, and survival are determined by the total protein content of the EVs from the plasma of melanomas and other solid tumors [131,193,194]. A detailed discussion about how EVs can be considered biomarkers in the context of different types of hematological malignancies appears below.

*AML*. EVs’ protein content in AML reflects the extent of the disease and correlates with the post-therapy likelihood of relapse. Low protein-containing EVs predict the long-term disease survival of AML. Moreover, a high level of TGF-β is observed in the EVs of AML patients upon diagnosis, which is dramatically reduced following chemotherapy, suggesting that EVs’ TGF-β levels mark the effect of chemotherapy in AML [189]. Moreover, EVs in the circulation of AML patients have been shown to be enriched with CD13, CD117, and CD34 [195,196]. EVs from AML cells also carry signature mRNA and miRNA molecules, thereby plausibly serving as future biomarkers [197].Signature mRNA molecules in the circulating EVs of AML patients often aid in the diagnosis and treatment of AML patients. For example, EVs’ FLT3-ITD and NPM1 mRNAs serve for determining AML prognosis; FLT3-ITD, IGF-IR, and CXCR4 mRNAs help in AML treatment, whereas IGF-IR, CXCR4, and MMP4 mRNAs aid in understanding the behavior of the tumor niche [197]. Moreover, miRNAs, such as miR-155, have been shown to be over-expressed in the serum EVs of AML patients compared to healthy individuals [198]. Therefore, AML-derived EVs could serve as diagnostic and prognostic biomarkers [131].

*CML*. CD13+ EVs were found to be up-regulated in the sera of CML patients compared to healthy individuals [195]. miR-215 is up-regulated in the plasma EVs of CML patients, and after successful treatment with imatinib, EVs’ miR-215 level is significantly reduced, serving as a predictor of successful treatment and discontinuing the drug [199].

*MM*. In the case of MM, plasma EVs, enriched with CD38, CD138, and superficially CD147, are considered to be biomarkers at different stages of the diseased condition [66,195,200]. For example, CD38+ EVs are associated with the clinical stages, and CD138+ EVs are correlated with disease stages and therapeutic response, whereas CD147+ EVs are related to the progression of MM. Moreover, CD44 enriched plasma EVs of MM patients are considered to be novel biomarkers for MM [201]. On the other hand, decreases in let-7b and miR-18a in the plasma EVs are associated with poor survival of MM patients [202].

*CLL*. Total EV populations in the plasma of CLL patients not only positively correlate with the advanced stages and overall patient survival but also with the duration of treatment in the initial stages of CLL [203]; these EVs are positive for CD19 and CD37. Another group reported that the level of plasma CD52+ EVs is increased in CLL patients and can be used as a predictive biomarker for CLL [139]. Furthermore, a moderate to high level of CD19, CD5, CD44, CD31, CD55, CD82, CD62L, HLA-A, B, C, and HLA-DR expression and a low level of CD49c, CD21, and CD63 expression was observed in CLL-EVs, which could be used as biomarkers for CLL [204]. A separate report indicated that EVs’ miR-155 could be considered a biomarker for patients suffering from B-cell CLL [205]. Chemotherapy-resistant individuals were found to high higher miR-155-laden EVs in the plasma, whereas their population was dramatically reduced in patients experiencing complete response [205].

*HL*. Patients with higher HL stages were shown to have a reduced level of CD30+ [a marker for Hodgkin and Reed–Sternberg (HRS) cells] EVs in the plasma [195]. The presence of miR-155, miR-127, miR21, and let-7 in the plasma EVs of HL patients often serves as a diagnostic tool for the detection and therapy of tumors in HL patients [198,206].

*WL*. In Waldenstrom macroglobulinemia (WM), a higher level of EVs and associated miR-155 was reported in the plasma compared to healthy controls, which could be used as a potential biomarker for WL [195,198]. However, EVs’ miRNA composition may change depending on the disease stages [207]. Table 4 summarizes how EVs act as a biomarker in different hematological malignancies.

*EVs in drug resistance*. Cancer cells often acquire resistance by decreased accumulation of drugs inside the cells, increased drug efflux outside the cells, compartmentalization of drugs, and alteration of cellular pathways targeted by specific drugs [208]. EVs have been shown to play an important role in modulating the drug resistance properties of various hematological neoplastic cells by various novel mechanisms. For example, BMSC-derived EVs confer resistance on MM cells against the chemotherapeutic drug bortezomib via the activation of diverse survival pathways, such as AKT, NOTCH1, NF-κB, and STAT3 [209]. Similarly, GAL3+ EVs from stromal cells confer anti-apoptosis and drug resistance to ALL cells by activating the NF-ĸB survival pathway [142]. In another study, BCL-derived EVs were found to express surface CD20, which helps BCLs to escape the humoral immune system by intercepting the chemotherapeutic drug rituximab [210]. The authors also indicated that sequestration of rituximab by EVs may contribute to the reduction in efficacy of pharmacological treatment [210]. Cancer cells often exert drug resistance properties by expelling the drugs into the extracellular space with the help of ATP binding cassette (ABC) transporter. ABCA3 transporters, highly expressed in leukemic cells, are localized on the exosome membrane within the MVBs, where drugs are effectively sequestered [210,211,212,213]. The release of exosomes is accompanied by the expulsion of drugs from the cancer cells, thereby contributing to drug resistance [210,214,215,216]. Furthermore, lymphoma cells, treated with chemotherapeutic drugs called anthracyclines, were shown to efflux the drug outside the cells through the release of EVs [217]. However, treatment of the cells with ABCA3 inhibitor increased the susceptibility of the drug via the suppression of EV biogenesis [217]. EV-mediated evasion of humoral immunotherapy was also reported to be dependent on ABCA3 in BCLs [210]. In the case of AML, exosomes from normal or AML bone marrow stromal cells (BMSCs) protect the leukemic cells owing to FLT3-ITD mutations against cytarabine; however, only AML-BMSC-exosomes impart protection against FLT3 inhibitor treatment [218]. Moreover, AML cells transfer their chemo-resistant properties to sensitive PML cells via the release of EVs [219]. 

In contrast to the above findings, EVs also effect potentiated chemotherapy-mediated elimination of cancer and other TME cells. For example, treatment with the anti-CD30 Ab drug conjugate Brentuximab Vedotin (SGN-35) in HL patients resulted in the binding of SGN-35 with HRS-derived CD30+ EVs and SGN-35/CD30+ EVs, not only killing the CD30+ tumor cells directly but also targeting CD30- cells in the TME, such as eosinophils and mast cells, resulting in severe damage to EV internalization [220]. Figure 5 describes how EVs influence chemotherapeutic treatment in the context of various hematological malignancies.

In summary, these studies indicate that EVs play a pivotal role in the regulation of hematological malignancies by modulating therapeutic resistance and tumor progression. Advancement of our understanding of cancer-associated EVs’ biology and novel techniques for isolating and characterizing the EVs will certainly help in predicting biomarkers for different hematological malignancies and in developing suitable targets.

## 7. Conclusions

A growing body of evidence indicates that EVs play a master role in cell–cell communication by delivering important messages among different types of cells. In recent years, the pace of EV research has been tremendous, and the ease at which EVs can be isolated with minimally invasive procedures has opened up possibilities for liquid biopsies and associated EVs to be used for real time monitoring of disease progression and treatment responses [221,222,223]. A recent multi-omics approach to longitudinally characterize EVs has been successful in elucidating hallmarks of COVID-19 throughout its progression [224]. Multi-omics studies have opened up avenues for quick, detailed characterization of EVs using new high-throughput techniques, as well as strict standardization, which could serve as an important benchmark for proper identification and could resolve potential problems arising due to the heterogeneity of EVs across patient population [225]. Some of the challenges could be overcome by utilizing EVs as an additional source of information when multiplexed with presently used biomarkers for better delineating disease mechanisms.

EVs have also shown promising results in drug delivery because of their inherent physiological stability and better immunological tolerance. They have been utilized to deliver different payloads ranging from onco-therapies to biologics and RNA therapeutics. Moreover, the growing body of evidence indicates the use of EVs as directed therapeutics or indirectly by stimulating immune cells [226,227]. Despite their therapeutic potential, several challenges remain to be overcome before the mass adoption of EV-based drugs. Roadblocks include several aspects of manufacturing, including heterogeneous production, which suffers from low yield and increasing complexity for scaling up production. On the bright side, there has been significant progress toward addressing these issues using microfluidic devices, along with bioreactors, to expedite production, as well as utilizing EV mimetics to address the problem of heterogeneity [228]. 

The findings reviewed here have raised new questions with plausible interpretations for understanding how cancer cells communicate with other hematologic cells in the TME and vice versa via the transfer of EVs. Moreover, EVs can be considered biomarkers for hematological malignancies, thereby helping not only in the diagnosis and prognosis but also in the development of therapeutic strategies against such malignancies.

Therefore, it is only a matter of time before the widespread adoption of EVs for multiple strategies. The increasing wealth of biological knowledge would only increase their potential applications, but future standardization practices across different fields must not be avoided to achieve a uniform approach to transforming EVs from bench to bedside applications.

## Figures and Tables

**Figure 1 biomolecules-13-00897-f001:**
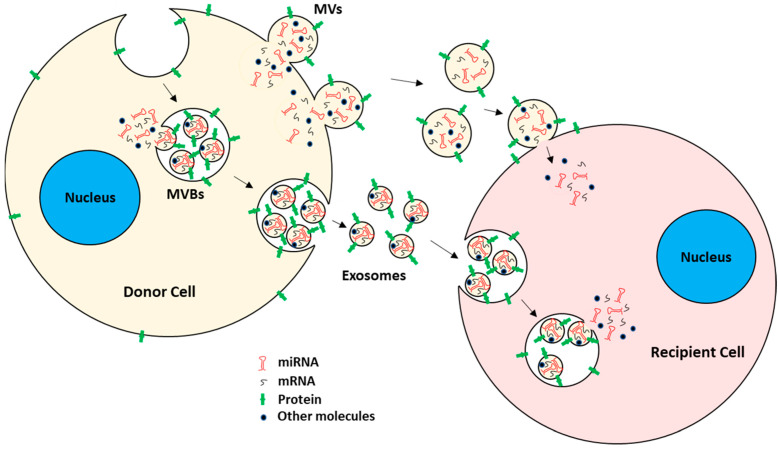
EV biogenesis and uptake by target recipient cells. EVs mainly consist of MVs and exosomes. MVs are generated by plasma membrane outward budding, whereas exosomes are of endocytic origin. Both carry mRNAs, miRNAs, proteins, and other bioactive molecules. EVs are taken up by recipient cells, either by direct fusion with the plasma membrane or by the endocytic pathway. After EV uptake, the EVs’ cargo is released into the recipient cells, hence acting as intercellular communicators between cells.

**Figure 2 biomolecules-13-00897-f002:**
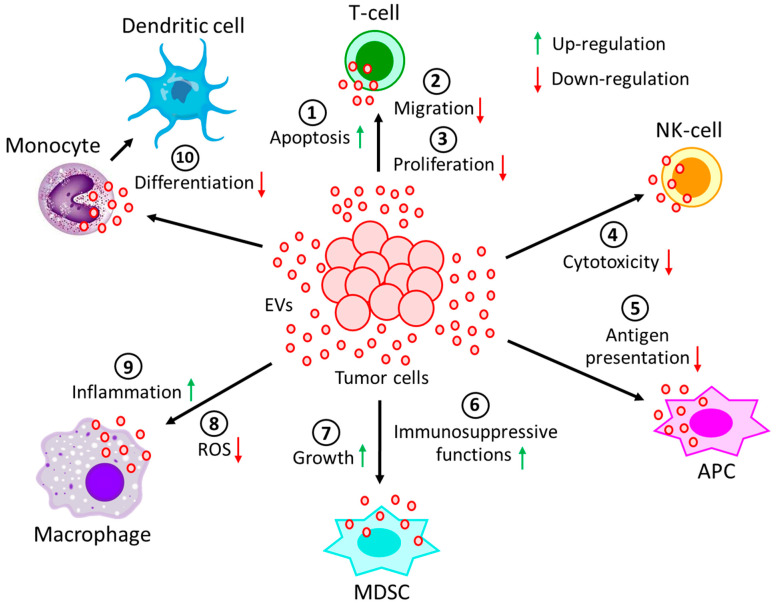
Effect of EVs generated during hematological malignancies on different immune cells. EVs, derived from tumor cells not only (**1**) induce apoptosis of T-cells, but also reduce T-cell (**2**) migration and (**3**) proliferation. Moreover, tumor-derived EVs (**4**) decrease the cytotoxicity of NK-cells and (**5**) restrain the processing of antigen by APCs. On the other hand, tumor-derived EVs (**6**) induce immunosuppressive functions of MDSCs, as well as (**7**) promote MDSCs growth. Again, tumor-derived EVs (**8**) not only prevent the generation of ROS in macrophages but also (**9**) promote macrophages’ pro-inflammatory response. (**10**) The differentiation of monocytes into dendritic cells is often perturbed by the incorporation of tumor-derived EVs into monocytes. All of these processes contribute to the development and progression of the tumor.

**Figure 3 biomolecules-13-00897-f003:**
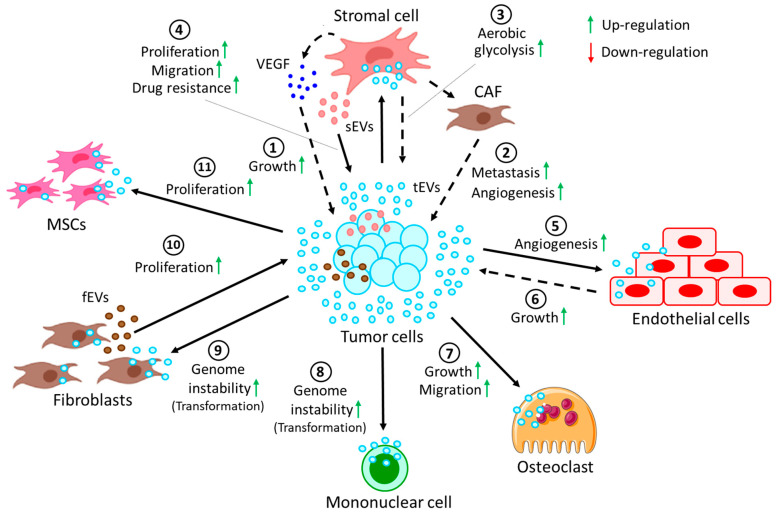
Crosstalk between tumor cells and cells in the TME through EVs. (**1**) EVs derived from tumor cells (tEVs) target stromal cells to release VEGF, which in turn induces growth of the tumor. tEVs induce the conversion of stromal cells into CAFs, which promote (**2**) tumor metastasis and angiogenesis. Moreover, tEV-fused stromal cells (**3**) induce the rate of aerobic glycolysis in tumor cells. On the other hand, (**4**) stromal cell-derived EVs (sEVs) promote proliferation and migration and confer drug resistance to tumor cells. (**5**) tEVs, upon fusion with endothelial cells, often promote angiogenesis, whereas (**6**) tEV-fused endothelial cells in turn contribute to tumor growth. (**7**) tEVs also promote the growth and migration of osteoclasts. More often, tEVs are shown to induce genomic instability in (**8**) several mononuclear cells and (**9**) fibroblasts, resulting in their transformation into pro-cancerous cells. (**10**) On the other hand, fibroblast-derived EVs (fEVs) induce the proliferation of the tumor. (**11**) Moreover, tEVs are also shown to induce the proliferation of MSCs, which in turn contribute to the progression of the tumor.

**Figure 4 biomolecules-13-00897-f004:**
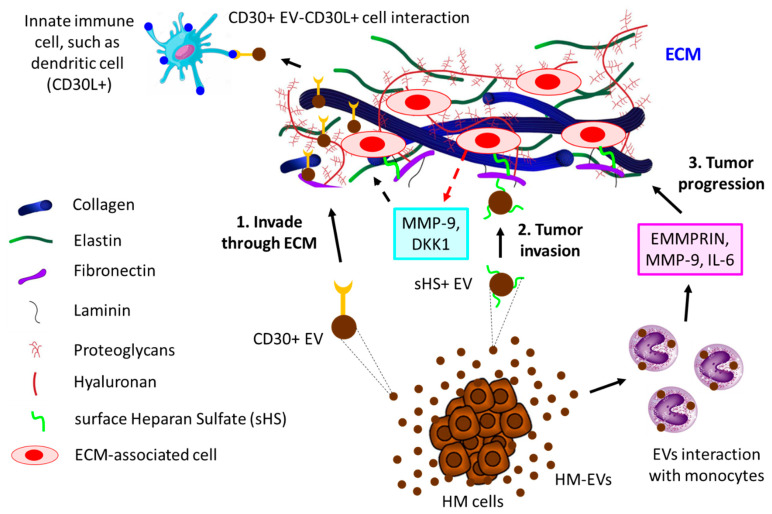
The interaction of HM-derived EVs with ECM leads to tumor progression. **1.** HL-derived CD30+ EVs invade through ECM to reach distant innate immune cells (here shown is a dendritic cell, for example) that are positive for CD30 ligand (CD30L+), leading to tumor cell–immune cell interaction. **2.** HM cells, such as MM-released sHS+ EVs, interact with ECM fibronectin, which further interacts with other ECM cells, thereby releasing MMP-9 and DKK1, which contribute to MM invasion through ECM. **3.** HC-derived EVs induce monocytes to secrete EMMPRIN, MMP-9, and IL-6, which not only promote tumor invasion but also influence tumor inflammation, ultimately leading to tumor progression through ECM.

**Figure 5 biomolecules-13-00897-f005:**
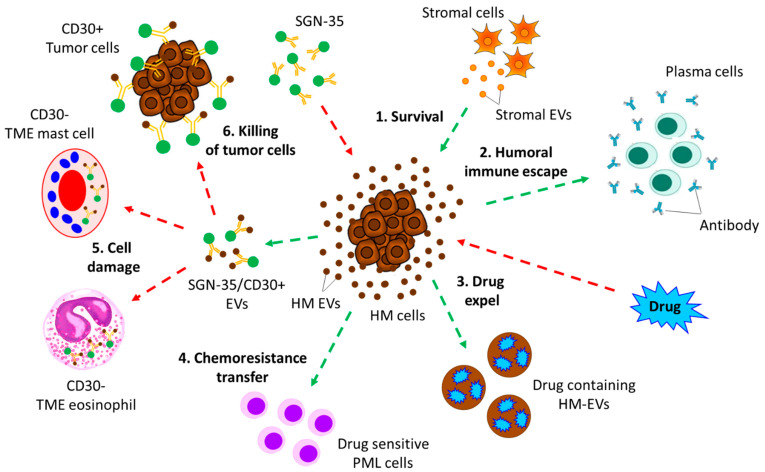
EVs influence chemotherapeutic drug treatment in various hematological malignancies. **1**. Stromal cells contributing to the survival of hematological malignant (HM) cells. Bone-marrow stromal cell (BMSCs)-derived EVs activate the survival pathways in MM cells; stromal cell-derived EVs also activate the survival pathway in ALL cells. **2**. EV-mediated humoral immune escape. BCL-derived EVs help BCL cells to escape humoral immune responses of the host. **3**. EV-mediated expulsion of chemotherapeutic drugs. Lymphoma cells expel the administered chemotherapeutic drugs, anthracyclines, via the release of EVs. **4**. EV-dependent chemoresistance transfer. Chemo-resistant AML cells transfer their chemo-resistant properties to chemo-sensitive PML cells via EV secretion. **5**. and **6**. HM-EV-mediated mast cell and eosinophil damage in the TME and killing of tumor cells. Treatment with an anti-CD30 bound drug conjugate, Brentuximab Vedotin (SGN-35), in HL patients results in the binding of SGN-35 with CD30+ HL-EVs, which in turn either is internalized by the CD30- eosinophils and mast cell in the TME, resulting in cellular damage, or directly kills CD30+ tumor cells. Green dotted arrow indicates the survival of the tumor, whereas red dotted arrow represents death or damage of the tumor cells.

**Table 1 biomolecules-13-00897-t001:** Classification of hematological neoplasms according to World Health Organization, 5th Edition [87,88].

Type	Sub-Class	Sub-Sub-Class	Characteristics
Myeloid neoplasm	MPN	CML	Excessive granulocyte proliferation leading to increased myeloid maturation
		Classic BCR:ABL1-negative MPN: (1) PV	Increased red blood cells
		(2) ET	Clonal disorder related to thrombocytosis
		(3) PF	Polyclonal increase in fibroblasts leading to BM fibrosis
		CNL	Overproduction of matured granulocytes
		CEL	Clonal proliferation of abnormal eosinophils resulting in hypereosinophilia
		Myeloproliferative neoplasm, unclassifiable	Does not have specific MPN characteristics or possess overlapping MPN features
	MDS	According to genetic abnormalities: (1) Low blasts and isolated del(5q)	5q deletion, BMB < 5% in BM, and PB < 2%, *SF3B1*/*TP53* mutation
		(2) Low blasts and *SF3B1* mutation	BMB < 5% and PB < 2%, *SF3B1* mutation, ≥15% ring sideroblasts
		(3) Biallelic *TP53* inactivation	BMB and PB < 20%; copy number loss/neutral results in loss of heterozygosity, complex cytogenetics
		According to morphology: (1) Low blasts	BMB < 5% and PB < 2%
		(2) Hypoblastic	BM cellularity < 20%
		(3) MDS-IB	MDS-IB1: BMB 5–9% and PB 2–4%; MDS-IB2; BMB 10–19% and PB 5–19%; MDS with increased blasts and fibrosis: BM fibrosis with BMB 5–19% and PB 2–19%
	AML	AML with recurring genetic abnormalities	Categorized into characteristic chromosomal rearrangements and no karyotypic abnormalities
		AML with myelodysplasia-related features	Lacks genetic features of AML but with history of MDS or MDS/MPN
		Therapy-related AML	Arises after chemo- or radiation therapy
		AML-NOS	With clinical, morphologic, and immunophenotypic features but lacking AML diagnostic karyotype
		Myeloid sarcoma	Tumor mass outside BM effaces into tissues
		Myeloid proliferation related to DS	DS, associated with increased AML
	Myeloid neoplasm with mutated *TP53*	MDS-mutated *TP53*	<10% blasts
		MDS/AML-mutated *TP53*	10–19% blasts
		AML-mutated *TP53*	>20% blasts
Lymphoid neoplasm	Precursor lymphoid neoplasm	B-cell ALL/LBL	Arises from B-cell precursor, representing most ALL/LBL
		T-cell ALL/LBL	Arises from T-cell precursor, representing 15% ALL/LBL
	Mature B-cell neoplasm	CLL/SLL	Small, mature-appearing lymphocytes
		LPL	Derived from post-germinal center of B-cells
		Monoclonal gammopathy	A type of lymphoid neoplasm for primary cold agglutinin disease
		Plasma cell neoplasm	Associated with terminally differentiated germinal centers of B-cells
		Hairy cell leukemia	Associated with post-germinal centers of B-cells
		MZL	Associated with marginal zone of matured B-cells
		FL: (1) Classic FL	Proliferation of centrocytes and centroblasts
		(2) FL grade 3B	Uncontrolled proliferation of centroblasts
		(3) FL-unusual feature	Comprising FL with blastoid features and FL with diffused growth pattern
		MCL	Neoplasm of naïve and antigen-presenting B-cells
		DLBCL	Neoplasm of matured B-cells
		High-grade B-cell lymphoma	Comprising aggressive BL and other high-grade B-cell lymphoma
	HL	cHL	Germinal and post-germinal centers of B-cells; again classified as nodular sclerosis cHL, mixed cellularity cHL, and lymphocyte rich- or lymphocyte-depleted cHL
		NLPHL	Associated with germinal neoplastic B-cells
	Matured T-cell/NK-cell lineage lymphoma	PTCL: (1) PTCL-NOS	Not fitting the criteria of T-cell lymphomas
		(2) ALCL	According to the expression of *ALK*, sub-categorized into “ALCL, ALK-positive” or “ALCL, ALK-negative” and “breast implant-associated ALCL.”
		(3) Follicular T_H_	Comprising AITL and related follicular T_H_ cell lymphomas
		(4) Extranodal	EBV-associated neoplasm arises in lymph nodes
		NK/T-cell lymphoma nasal type	
		(5) Hepatosplenic T-cell lymphoma	Aggressive and related to immunosuppression
		(6) Primary intestinal T-cell lymphoma	Aggressive T-cell lymphoma in intestinal tract
		Primary cutaneous T-cell lymphoma	Lymphoma of cutaneous peripheral T-cells
		ATL	Associated with peripheral T-cells from HTLV-1 infected CD4^+^ T-cells
		TLGL leukemia	Derived from clonally expanded granular T-cells
		T-cell polymorphocytic leukemia	Highly aggressive, medium-sized matured T-cells
		NKLGL leukemia	Aggressive, associated with malignant NK-cells
		Aggressive	Aggressive, associated with malignant NK-cells
		Aggressive	Nasal type, associated with EBV infection
		NK-cell leukemia	
Myeloid and lymphoid neoplasm with eosinophilia and *TK* gene fusion	HES	Primary (or neoplastic)	Clonal eosinophilic expansion reaching underlying stem cells and myeloid and eosinophilic neoplasm
		Secondary (or reactive)	Polyclonal eosinophilic expansion mediated by overproduction of eosinophilic cytokines during parasitic infections; certain solid tumor T-cell lymphomas
		Idiopathic	The underlying cause of HE remains unknown
	Specific syndrome associated with HE	-	Disease complications and clinical presentation are not fully defined, e.g., EGPA and some immunodeficiencies
	HE_US_	-	Persistent unexplained HE; difficult to predict whether the patients will develop clinical manifestations leading to HES
Mastocytosis	SM	Indolent SM	70% of SM is found to be indolent SM, which may or may not cause skin lesions of maculopapular cutaneous mastocytosis (MPCM)
		Smoldering SM	Rare, shows same phenotypic effects as indolent SM
		Aggressive SM	Highly aggressive, manifesting round, rather than spindle-shaped, mast cells with median survival from month to years
		MCL	Same phenotypic responses as aggressive SM
		SM-AHN	Similar to SM phenotype but requiring urgent treatment depending on the disease stage in BM and extracutaneous sites
Histiocytic or dendritic neoplasm	Histiocytic sarcoma	-	Malignant proliferation of cells with morphologic and immunophenotypic features of mature tissue histiocytes
	Tumors of Langerhans cells	Langerhans cell histiocytosis	Malignancies of cells expressing CD1a, langerin, and S100
		Langerhans cell sarcoma	High-grade malignancy with same features as Langerhans cell histiocytosis
	Indeterminate dendritic cell tumor	-	Rare, involving proliferation of cells, spindle-shaped to ovoid, similar to IDC
	Interdigitating dendritic cell sarcoma		Very rare, involving spindle to ovoid cells resembling interdigitating dendritic cells
	Follicular dendritic cell sarcoma		Neoplastic proliferation of spindle to ovoid cells; morphologically and immunophenotypically similar to follicular dendritic cells
	Inflammatory pseudotumor-like follicular/fibroblastic dendritic cell sarcoma		Neoplastic spindle cells residing in lymphoplasmacytic infiltrate involving liver and spleen
	Fibroblastic reticular cell tumor		Very rare, spindle cells with cytokeratin involving skin, spleen, and lymph nodes
	Disseminated juvenile xanthogranuloma		Proliferation and dissemination of small oval histiocytes resembling dermal juvenile xanthogranuloma of skin and soft tissues
	Erdheim–Chester disease		Involving foamy histiocytes of bones leading to clonal proliferation and associated with heart, CNS, and retroperitoneum
Mixed myeloid and lymphoid neoplasm	MPAL	Ph^+^ MPAL	With Philadelphia chromosome (Ph)
		Ph^−^ categories: (1) MPAL with t(v;11q23.3); KMT2A-rearranged	MPAL associated with detectable t(v;11q23.3) and *KMT2A* rearrangement
		(2) MPAL B/myeloid NOS	MPAL with B-lymphoblast immunophenotype lacking Ph chromosome or t(v;11q23.3)
		(3) MPAL T/myeloid, NOS	MPAL with T-lymphoblast immunophenotype lacking Ph chromosome or t(v;11q23.3)

*Abbreviations not provided in the text nor in the table:* BCR, B-cell receptor; ABL1, Abelson murine leukemia viral oncogene homolog 1; SF3B1, splicing factor 3b subunit 1; TP53, tumor protein p53; EGPA, eosinophilic granulomatosis with polyangiitis; HE_US_, hypereosinophilia of undetermined origin; SM-AHN, systemic mastocytosis with an associated hematological neoplasm; IDC, indeterminate cells; CNS, central nervous system; KMT2A, lysine methyltransferase 2A.

**Table 4 biomolecules-13-00897-t004:** EVs as biomarkers of different types of hematological malignancies.

Disease	EVs’ Biomarker	Up-/Down-Regulation	References
AML	TGF-β	Up	[189]
	CD13, CD117, CD34	Up	[195,196]
	FLT-ITD, NPM1, IGF-IR, CXCR4, MMP4 mRNA	-	[197]
	miR-155	Up	[198]
CML	CD13	Up	[195]
	miR-215	Up	[199]
MM	CD38, CD138, CD147	Up	[66,195,200]
	CD44	Up	[201]
	let-7b, miR-18a	Down	[202]
CLL	CD19, CD37	Up	[203]
	CD52	Up	[139]
	CD19, CD5, CD44, CD31, CD55, CD82, CD62L, HLA-A, B, C, HLA-DR	Up	[204]
	CD49c, CD21, CD63	Down	[204]
	miR-155	Up	[205]
HL	CD30	Down	[195]
	miR-155, miR-127, miR-21, let-7	Up	[195,206]
WL	miR-155	Up	[195,207]

## Data Availability

Not applicable.

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
