# Peer review of "The Role of Extracellular Vesicles in the Pathogenesis of Hematological Malignancies: Interaction with Tumor Microenvironment; a Potential Biomarker and Targeted Therapy"

_biomolecules, 2023, doi:10.3390/biom13060897_

Round 1
Reviewer 1 Report
The aim of this review was to comprehensively investigate the interactions between extracellular vesicles released by tumor cells and the tumor microenvironment. The authors conducted a significant analysis of the literature, a large number of sources were used in the work. The topic of the review is interesting, although highly specialized, but this review can be very useful for specialists in the field of hematology.
Despite the advantages of this work described above, I would like to note its special points that need to be completed.
1. Firstly, the article is devoted to hematological malignancies, but the work completely lacks information about this type of neoplasm. This is a scientific review and should be understandable to a wide range of readers. In this regard, I propose to add a section in which to give a classification of hematological neoplasms. Perhaps this section would be appropriate in the form of a table (clearly structured in accordance with the modern classification) containing the main types of hematological malignancies and their brief characteristics, with a little description in the text. The authors do not differentiate information between individual types of hematological neoplasms, but this can have a particular importance.
2. Secondly, the rationale for choosing hematological neoplasms as the main object of review is missing from the text. This may be a very short paragraph, but it should explain the logic of the authors choice.
3. The text also lacks a separate section on the interaction of EVs and endothelial cells. Moreover, the absence of any mention of the lymphatic system is especially unusual. Lymphatic vessels are an important part of the tumor microenvironment, as are blood vessels, immune and stromal cells, and the extracellular matrix. Some hematological neoplasms have the lymphoid origin and spread through the lymphatic vessels, therefore, the data regarding endothelium of the blood vessels, as well as the lymphatic endothelium should be provided.
4. Finally, the references list contains a single number of studies after 2019, it may be necessary to revise the sources used, and add more recent references.
Reviewer 2 Report
The authors provided a well-documented overview of Ev's role in the context of hematological malignancies. In addition to this, the review could represent a really interesting point of view in a field so dynamic and rich in potential future applications. If the article is well written, some points should be fixed:
1. Paragraph "Tumor-derived EVs":
First line: Tumor-derived EVs are distinguished from normal cell-secreted EVs due to the presence of unique tumor-specific ‘labels’. ADD REFERENCES (Es. PMID: 35141731, PMID: 37012233, PMID: 36310768 or others)
2. Font size of the different tables needs to be harmonized.
3. Paragraph "EVs: The biomarker of hematological malignancies: Contribution in drug resistance" seem to be an "orphan" section: authors should decide if make it more informative or delete it.
4. Conclusions are too poor in terms of future perspectives, challenges, and opportunities.
I hope that my comments could be useful, and I look forward to reading the revised version of the paper.
Good luck.
Good Enghish
Reviewer 3 Report
The manuscript provides a comprehensive review to the involvement of extracellular vesicles (EVs) in hematological malignancies including the modulation of immune cells' functions, the communications between tumor cells and stromal cells, the interaction to extracellular matrix, and for biomarkers and drug resistance. Some revisions may consider.
1. For ECM and drug resistance, the authors could also design a summarized figure.
2. It is better to include a prospective/future direction part to suggest the most important works for this field that are required to be solved.
Round 2
Reviewer 1 Report
The authors have significantly revised the manuscript in accordance with the suggested comments, and I believe that this has seriously improved the quality of the manuscript.
Author Response
Thanks for your review.